# Recent Advancements in the Recovery and Reuse of Organic Solvents Using Novel Nanomaterial-Based Membranes for Renewable Energy Applications

**DOI:** 10.3390/membranes13010108

**Published:** 2023-01-13

**Authors:** Indrani Gupta, Oindrila Gupta

**Affiliations:** 1Department of Chemistry and Environmental Science, New Jersey Institute of Technology, Newark, NJ 07102, USA; 2Vertex Pharmaceuticals Inc., Boston, MA 02210, USA

**Keywords:** biofuel, separation processes, membrane technologies, organic solvent separation, nanomaterials

## Abstract

The energy crisis in the world is increasing rapidly owing to the shortage of fossil fuel reserves. Climate change and an increase in global warming necessitates a change in focus from petroleum-based fuels to renewable fuels such as biofuels. The remodeling of existing separation processes using various nanomaterials is of a growing interest to industrial separation methods. Recently, the design of membrane technologies has been the most focused research area concerning fermentation broth to enhance performance efficiency, while recovering those byproducts to be used as value added fuels. Specifically, the use of novel nano material membranes, which brings about a selective permeation of the byproducts, such as organic solvent, from the fermentation broth, positively affects the fermentation kinetics by eliminating the issue of product inhibition. In this review, which and how membrane-based technologies using novel materials can improve the separation performance of organic solvents is considered. In particular, technical approaches suggested in previous studies are discussed with the goal of emphasizing benefits and problems faced in order to direct research towards an optimized membrane separation performance for renewable fuel production on a commercial scale.

## 1. Introduction

The environmental concerns over the restricted supply of crude oil and the excessive surge in oil prices in recent years have rekindled an interest in the invention of technologies that promote the use of renewable resources to produce energy. Recently, among many alternative sources of energy, biofuel is a promising renewable resource for strengthening energy sustainability and security and reducing the emission of greenhouse gases. Fermentation is a potential alternative for production of chemicals and fuels from renewable resource. However, fermentative products are usually present at low concentrations and conventional separation methods are highly energy intensive and produces a lot of side reactions. Moreover, microorganisms are generally deactivated by the presence of organic solvents, resulting in decreased fermentation productivity [1]. Therefore, the development of economical processes for product recovery from fermentation broth is necessary to meet the energy demands worldwide, without relying on fossil fuel reserves, thus reducing global warming. A major challenge behind the production and application of biofuel is the availability of effective separation and technologies, which account for at least 40–80% of the entire process cost [1].

To prevent solvent inhibition, many technologies, such as adsorption, perstraction, gas stripping, and liquid–liquid extraction, have been developed as a continuous mode in combination with fermentation systems [2]. Each of these technologies have their own drawbacks, such as unwanted removal of nutrients in adsorption, addition of toxic solvents into fermentation broth by perstraction and extraction, and decreased ethanol removal rate by gas stripping. Membrane-mediated separation technology started gaining significant popularity in the latter half of 1960s [3]. On contrary to the conventional methods, such as energy intensive distillation and evaporation, membrane separation has several advantages, such as less energy consumption, high separation factors, and small carbon footprint [4,5]. Table 1 lists the advantages and disadvantages of various solvent recovery processes. Membranes have mostly been used in gas separation and water purification [6,7,8,9,10,11]. However, up until now, using conventional membranes to separate various organic solvents from fermentation broth remains a challenge, mainly due to the chemical instability of conventional polymeric membranes in organic solvents as well as the high solubility of these organic solvents in water [12,13]. Yet, due to the high demand of increasingly growing petrochemical, pharmaceutical, and catalyst industries the separation of organic solvents is remarkably becoming more crucial [14,15], as well as the need to shift the focus from fossil fuel reserves to renewable fuels [16]. Hence, there is an immediate need for the development of novel membranes to compensate for this technological gap and to improve the membrane capacity for organic solvent separation. This review article focusses on various nanomaterial-based membranes for solvent separation for renewable energy production. 

## 2. Membrane Materials for Organic Solvent Recovery

To obtain enhanced performance in terms of flux and selectivity, most solvent-selective membranes are polydimethylsiloxane (PDMS), poly[1-(trimethylsilyl)-1-propyne] (PTMSP), and polyvinylidene PVDF [31,32,33]. Mixed matrix membranes (MMMs) have emerged as suitable candidates for applications in membrane-mediated separations, such as carbon capture [34], desalination [35], and pervaporation [36]. Typically, MMMs consist of two phases: a dispersed phase and a continuous polymeric phase [36,37]. The combination of two phases give rise to superior qualities and functionalities in developing a membrane. The dispersed phase plays a role in molecule-selective property, while the continuous polymeric phase demonstrates enhanced processability for an active layer that is thin and defect-free. The addition of inorganic fillers can also help in enhancing the properties of permeation by modifying the chain packing and increasing the free volume that exists in the polymer matrix [38,39]. For obtaining a high selectivity, the pores act as molecular sieves for desirable separation. Thus, the performance of MMMs in terms of separation overcomes the trade-off that exists in polymer membranes in terms of permeability and selectivity.

Although a lot of membrane materials have been examined for the purpose of recovering organic solvents from diluted fermentation broths, the materials with excellent solvent-permselective qualities are few and their separation performances fail to meet the demands of commercialization at this stage. Figure 1 represents the publication trend of articles that fabricated membranes for organic solvent separation. In this review, the separation performances of the solvent-permselective membranes using novel nanomaterials are summarized. An overview on the analytical aspect of these solvent-permselective nanomembranes, including their challenges and opportunities, is also presented.

### 2.1. Graphene Oxide Quantum Dots

GOQD is a promising class of nanomaterials that is increasingly gaining importance in the field of biomedical applications [40,41], sensors [42], and electronics [43] owing to its superior biocompatibility and excellent optical properties. GOQDs consists of layers of graphene sheets whose lateral size is less than 100 nm and have the same sp^2^ carbon structure and hydrophilic functional groups as graphene oxide (GO) but is much smaller than GO. This provides a shorter pathway for molecules to permeate. Quantum dots have been used for different separation processes, such as nanofiltration [44] and forward osmosis [45]. Owing to its hydrophilic character, GOQDs within the polymer matrix are expected to adsorb water molecules and enhance the alcohol/water mixture separation.

Wang et al. [46] developed a novel membrane for ethanol dehydration by combining GOQDs with sodium alginate (SA). GOQDs that had an average lateral dimension of around 3.9 nm enabled shorter and simpler transport pathways for water molecules as opposed to microsized GO sheets. A 60% enhancement in ethanol dehydration flux with a separation factor of 1152 ± 48 was observed for the GOQDs (cumulative permeate flux: 2432 ± 58 g m^−2^ h^−1^) as compared to that of pristine alginate membrane and it is attributed to the high hydrophilicity and nano size of the GOQDs. A recent article incorporated graphene oxide quantum dots (GOQD) in a polyvinyl alcohol (PVA) matrix for investigating the dehydration performance of alcohol/water mixtures [47]. When 300 ppm of GOQD (PVAx_GOQD300) was incorporated on the membrane, the highest separation factor (476.4 ± 8.25) and PSI value (2.20 × 105 g m^−2^ h^−1^) were observed. A lesser separation performance was observed for linear alcohol for the PVAx-GOQD300 as compared with the sterically hindered alcohol as seen in Figure 2.

In another study [48], Graphene quantum dots (GQDs) were employed to cover the structural defects on the graphitic sections of reduced graphene oxide(rGO). After successfully covering the structural defects, a dense composite membrane that was built from the nanocomposite filler (rGO + GQD) incorporated into an alginate solution efficiently separated alcohol/water and exhibited a good separation performance of alcohols that have a lower molecular weight. Figure 3 shows the blockage of alcohol molecules and successful permeation of water molecules after sealing and curing the structural defects on rGO with GQDs. The best pervaporation performance in the separation of methanol/water mixture with a permeate flux of 2323 g m^−2^ h^−1^ and water concentration in permeate of 92.7% at 70 °C was observed for alginate that was combined with 3 wt% of rGO + GQD(Alg_rGO + GQD), as shown in Figure 4.

A study led by Wu et al. [49] prepared ultra-thin PA layer by the direct interfacial polymerization (IP) on a ceramic substrate that was macroporous in nature. The ample hydroxyl, carboxyl, and amino groups present in aqueous phase additives provided attachment spots for ethylenediamine (EDA) to develop an improved membrane structure. The TFN PV membrane exhibited a very high separation performance for the dehydration of ethanol. Amino-modified carbon quantum dots provided more rapid penetration channels for water. The large pore size (~300 nm) of macroporous ceramic substrate posed a challenge to direct IP. The aqueous phase tended to easily penetrate the pores, which led to an incomplete formation of PA layer as shown in Figure 5. There should be an even distribution of the aqueous liquid layer on the surface to make it defect free. However, it was observed that only at a relatively high concentration of PVA was a defect-free layer formed which affected the permeate flux. A network structure with lesser PVA concentrations was formed by N-CDs to lower the diffusion rate of EDA in order to build a very thin PA layer as shown in Figure 5.

In another article, the dehydration of 90/10 wt% butanol/water mixtures was carried out using a nanocomposite TFN membrane composed of tannic acid (TA), acyl chloride monomers (TPC or TMC), and nitrogen-doped graphene quantum dots (NGQD) [50]. The role of the NGQD was to block the permeation of larger alcohol molecules and increase the membrane affinity to water molecules. The alcohol dehydration capability of TFN membrane was also experimented with using other alcohols, such as butanol isomers, and it was discovered that the permeation rate was faster for those alcohols that have a more linear and less sterically hindered structure, as shown in Figure 6.

### 2.2. Zeolitic Imidazolate Frameworks (ZIFs)

Zeolitic imidazolate frameworks (ZIFs) are a relatively new class of metal organic frameworks composed of metal ions and organic imidazolate linkers, with a similar structure like conventional aluminosilicate zeolites. Their highly porous nature, excellent chemical and thermal stabilities, and multiple functionalities have caused ZIF materials to be used for several different applications. Particularly, due to the hydrophobic nature of ZIF-8, it has gained attention for high alcohol separation [51].

A recent article showed that ZIF-8-based mixed matrix membrane (MMM) exhibits weak stability in the separation of acetone-butanol-ethanol (ABE) from a fermentation broth and it is attributed to the degradation of the ZIF-8 framework structure caused by organic acids. In order to increase the stability, ZIF-8-derived nanoporous carbon (ZNC) was developed through a direct carbonization process and it replaced ZIF-8 in MMM [52]. The ZNC-based MMM exhibited good compatibility and high hydrophobicity between particles and polymer. It was interesting to see that during the 100 h of continuous processing of ABE from fermentation broth, a steady performance was observed by the ZNC incorporated MMM with an average cumulative flux of 1870 g m^−2^ h^−1^ and a good separation factor of 20 for n-butanol.

Xu et al. reported the development of novel PDMS/DLA-ZIF-90 MMMs by bringing about a hydrophobic modification of ZIF particles through dodecylamine (DLA) for pervaporation recovery of organic solvents for the first time [53]. For DLA modification, the already developed ZIF-90 particles (DLA-ZIF-90) were mixed in 10 wt% DLA/MeOH solution followed by a reflux at 60 °C for 30 min. The DLA-ZIF-90 particles were centrifuged at 10,000 rpm for 10 min followed by a methanol wash and was dried for 24 h in a vacuum oven at 80 °C. The reaction mechanism is shown in Figure 7, where DLA was attached to the ZIF-90 via imine condensation. The authors concluded that there may exist non-selective interfacial void regions between unmodified ZIF particles along with PDMS chains as shown in Figure 8. There is a higher compatibility between hydrophobic dodecyl chains attached on ZIF-90-DLA particles and hydrophobic PDMS chain segments, thus enhancing the bonding of ZIF-90 onto the PDMS matrix and removing the interfacial void to build the filler–polymer interface. Results showed that PDMS/DLA-ZIF-90 mixed matrix membrane exhibited higher separation factor and permeate flux as compared to other PDMS-based membranes in the literature, which is attributed to the flexibility of the inner channels of DLA-ZIF-90 particles which enhances adsorption selectivity along with the affinity between ZIF-90-DLA particles and PDMS matrix.

A study led by Pan et al. modified the superhydrophobic form of zeolitic imidazolate framework-8 (ZIF-8) layer with a nanosized, bud-like exterior morphology by n-octadecylphosphonic acid modification for the efficient separation of ethanol/water mixture, by making the surface superhydrophobic and attracting ethanol molecules [54]. The highest separation factor (17.4) and a comparable flux was recorded for the optimal membrane. Long term studies also proved its high operational stability. In another study, MMMs were prepared by doping nanosheets of 2D ZIF-L with porous frameworks into an oleophilic matrix to enrich ABE from the modeled fermentation broth. The modified membrane showed an improved flux over polymeric membranes (72.3%) and separation factor (106%) of organics over water with only 5.0 wt% of ZIF-L fillers incorporated into the matrix. ZIF-L based MMMs decreased the permeate water content below 30 wt%, which is needed for ABE utilization in the real bio-refinery industry.

A study led by Zhu et al. incorporated ZIF-8 nanoparticles onto the graphene oxide (GO) nanosheet surfaces and filled them into polydimethylsiloxane (PDMS) matrix for ethanol separation [55]. ZIF-8 combined with GO composites exhibited excellent compatibility with PDMS along with superior dispersion in PDMS matrix over ZIF-8 nanoparticles as a stand-alone material. A cumulative flux of 443.8 g/m^2^ h with 5 wt.% ethanol aqueous solution at 40 °C and a separation factor of 22.2 was obtained. GO nanosheets combined with hydrophobic ZIF-8 nanoparticles behaved as a strong barrier for the pervaporation recovery of ethanol with the transport of feed solution occurring via the continuous inner channels of ZIF-8 on the GO surface and PDMS matrix, as shown in Figure 9.

In a similar study, ZIF-8 modified graphene oxide (ZGO) with a polyether block amide (PEBA) group was incorporated on a ceramic based tubular substrate in order to develop composite membranes for the recovery of bio-butanol. ZGO laminates formed channels of transport for the enhanced permeation of butanol molecules [56]. The enhancement in permeate flux and separation factor of the composite membrane raised the mass transfer of the ZGO laminate. During the separation of 5% butanol from its aqueous solution at 55 °C, the permeate flux by the ZGO/PEBA composite membrane was 1001 g m^−2^ h^−1^ with a separation factor of 29.3, indicating its high potential to recover biobutanol from an aqueous medium.

### 2.3. Carbon Nanotubes

Carbon nanotubes (CNTs) have garnered remarkable attention as a novel type of nanofillers because of their extraordinary structures and properties [57]. CNTs appear as rolled-up cylinders of graphite sheets, with a tubular morphology and having a diameter in the nanometer range, that are held together by strong van der Waals attraction [58]. Thus, the effectiveness of using CNTs in separation studies depends on its ability to form a uniform dispersion in the polymer matrix. The attachment of functional groups to the surface of CNTs is considered an effective strategy to inhibit aggregation [59], and treating surfaces with strong acids, such as nitric and sulfuric acid, is conventional for this purpose [60].

Electrospun nanofiber membranes (ENMs) have gained importance in membrane distillation (MD) processes due to their highly porous structures and open pores that are interconnected. A recent article presented a facile fabrication technique for the surface coating of superhydrophobic ENMs via a spraying method [61]. The agglomeration of CNTs was prevented using a lower CNT content and a suitable solvent for dispersion. Owing to their low surface energy and high hydrophobicity, the fabricated membranes exhibited superhydrophobic property and exhibited a high-water flux of 28.4 kg/m^2^ h with a steady vacuum membrane distillation (VMD) performance greater than 26 h.

Yang et al. [62] developed open-ended aligned CNT/(polydimethylsiloxane) PDMS membranes which appeared like a hamburger with nano-channels (∼10 nm) in the middle layer for applications in ultrafiltration and angstrom cavities in the embedded PDMS for use in pervaporation. The aligned CNT membranes overcome the limitation of filling content of the nonaligned CNT/PDMS membrane, bringing about great mechanical properties and an enhancement in selectivity and mass flux for alcohol separation. The membranes break the tradeoff between permeability and selectivity with both the parameters significantly increasing for alcohol separation. The authors showed that penetrant molecules selectively permeate through the internal nanochannels of CNT with increasing membrane permeability, and thus generate the idea to develop a design for highly efficient nano channeled membranes for organic solvent separation.

The size of the transient gaps or pores for penetrant molecules in the PDMS matrix are usually several angstroms, thus strictly restricting the molecular diffusion. As the size of penetrant molecules increased, the diffusion coefficients decreased. As depicted in Figure 10, the average diameter of the open-ended CNTs that serve as orderly nanochannels is 9.67 nm, which allows for the successful diffusion of water and butanol with several angstroms through the membrane. 

Gupta et al. [63] demonstrated ethanol separation from its aqueous mixture through microwave-induced sweep gas membrane distillation using carbon nanotube immobilized membranes (CNIM). The synergistic effect of CNIM coupled with microwave heating was most effective where the ethanol permeate flux was found to be 11.3 L/m^2^ h with a separation factor of 13.7, which were 46% and 102% greater than that of conventional membrane distillation. In another study led by Gupta et al. [64], separation of acetone, butanol, and ethanol (ABE) mixture from dilute aqueous fermentation products was performed using Carbon nanotubes (CNTs) and octadecyl amide (ODA) functionalized CNTs. An enhancement in the ABE flux was observed to be as high as 105%, 100%, and 375% for the CNIM and 63%, 62%, and 175% for CNIM-ODA respectively. The mechanism of action behind enhanced separation was due to CNT immobilization on the active membrane layer which caused the preferential selective sorption of ABE and was validated by a reduction in contact angle measurements.

Xue et al. [65] prepared carbon nanotubes (CNTs) filled polydimethylsiloxane (PDMS) hybrid membrane for recovery of butanol from ABE fermentation broth. Due to the selective sorption sites of the CNTs with super hydrophobicity, the mass transport through the smooth surface and the inner walls increased significantly. The highest cumulative flux was observed at 244.3 g/m^2^·h with a butanol separation factor of 32.9, with 10 wt% CNTs in the PDMS membrane and at 80 °C. This indicated that the CNT/PDMS membranes have a great potential for the separation of butanol from ABE fermentation broth through pervaporation.

In another study, carbon nanotube (CNT)-mixed polydimethylsiloxane (PDMS) membranes were utilized for the recovery of ethanol from model solutions and for fermentation by yeast that is self-flocculating in nature [66]. The time course for the fed batch fermentation process of ethanol using CNT-mixed membranes is shown in Figure 11. Ethanol fermentation started with ∼240.0 g/L of glucose and ethanol and other by-products were formed gradually over time. Product recovery in situ along with the CNT-mixed membrane regulated the product titer inside the fermentation broth within a stable range by continuous removal of ethanol. The authors concluded that CNT-mixed membrane combined with ethanol fermentation by self-flocculating yeast decreases ethanol-mediated cell inhibition and lowers the cost of production due to decreased fouling.

### 2.4. Graphene Oxide Membranes

Graphene oxide (GO) has achieved immense interest in the fields of separation science due to the presence of hydrophilic functional groups, e.g., carboxyl, hydroxyl, and epoxide groups [67]. GO sheets allow uninterrupted permeation of water vapor while completely preventing the permeation of other gas molecules [68,69,70]. Although there is a heightened interest in water transport through the layers of GO membranes, there exist limitations in the membrane fabrication steps when the real-world water separation challenges come up. Shin et al. [71] prepared a polyethersulfone (PES)-supported GO membrane for the separation of ethanol/water at different temperatures. The molecular transport of water–ethanol mixtures is negligible for graphene bilayer spacing below 1 nm, as confirmed by molecular dynamics simulation. The authors suggested that the permeation of water and ethanol into the GO interlayers at its earliest stage may be the most important step in controlling and tuning the selectivity. In a recent study led by Munoz et al. [72], hydrophilic GO at various concentrations was incorporated into PVA matrix for ethanol dehydration by pervaporation (PV), showing a 75% enhancement over cross-linked PVA membrane. Their results showed that the best performance was obtained at 1 wt% GO which exhibited a PV flux of 0.14 kg m^−2^ h^−1^ and a separation factor of 263. Figure 12 shows the effect on GO content based on different operating temperatures on permeate flux. The cumulative permeation rate increased with a double increase of GO loading which was attributed to an increase in free volume. On the other hand, the permeation rates were increased by preferential adsorption of the higher polar compound (water) due to the highly hydrophilic nature of GO.

Liu et al. [73] used molecular dynamics (MD) simulations to understand the mechanism of water–ethanol separation through the monolayer of GO membranes with varying pore sizes and O/C ratios. The separation properties under 1:1 water–ethanol mixtures revealed that water selectivity was favored with a higher O/C ratio, membrane pore size, and water flux (Figure 13). Due to oxidization, the sorption of water rules the permeation process. It was interesting to note that, alternatively, water diffusion leads to improved permeation at low oxidization conditions (Figure 14). Their study showed the importance of pore diameter and O/C ratio on the microscopic level in the separation of water–ethanol through the pores of GO membranes and uncovered the ruling effects for the permeation of water with the GO material serving as a potential alternative in water–ethanol separation technologies.

A recent article by Talyzin et al. [74] showed that GO membranes present in liquid solvents that are solvated and hydrated significantly differ from GO precursor powders. Only one ethanol layer is sufficient to hydrate the GO membranes and the composition remains unchanged upon cooling to around 140 K. On the contrary, at low temperatures in ethanol, phase transitions are exhibited by GO powder into a two-monolayer solvate whereas H-GO shows “pseudo-negative thermal expansion” with up to four inserted alcohol monolayers at low temperatures. Both hydrated membranes form a liquid-like monolayer which is believed to be responsible for fast water permeation through GO membranes. The presence of ethanol in water causes a smaller d (001) spacing inside the membrane lattice and an opposite trend is observed for GO powders which is why membranes are not permeated by ethanol. The geometry of the GO sheet edges determines the count of ethanol layers that would be intercalated into the layered structure of GO. The geometry of the membranes contains edge groups present over the functional groups of neighboring GO planes, and the powder has an edge over edge configuration of functional groups, as shown in Figure 15. The authors believed the GO membranes exhibited a quick exchange of water molecules between the liquid media and the lattice, while its entry to interlayers is blocked by ethanol, which is highly important in understanding membrane deposition and the permeation of liquid mixtures through the membrane matrix. Liu et al. [75] performed molecular simulations to understand the water–ethanol permeation through the utilization of a novel type of Janus GO membranes with variable orientations of pristine and oxidized surfaces. Their results showed that the GO membrane was endowed by the oxidized upper surface with a good capability to capture water and the effective vertical diffusion of water molecules was also promoted due to the in-built oxidized interlayer.

The commercial use of GO membranes continues to be challenging for the molecule separation process with strong coupling effect and less size inconsistency such as water–ethanol. In a very recent article, a new exclusive technique of constructing quick water channels in GO membrane was demonstrated for enhanced water–ethanol separation combining the synergistic effects between hydrophilic polyelectrolyte (polyethylenimine) and zwitterion-functionalized GO, as shown in Figure 16 [76]. The built in ordered and steady channels contained ionic hydrophilic groups with a high density, which plays a role in negating the strong coupling force between ethanol and water, allowing the water molecules to quickly permeate and restricting the transport of ethanol molecules. A very high separation factor (2248) and a flux of 3.23 kg/m^2^ h for separating water–ethanol mixture has been reported which led to the construction of 2D channels for the efficient separation of strong-coupling mixtures.

Yeh et al. [77] developed a multilayered membrane for the pervaporation of ethanol dehydration by laminating a GO layer that acted as a barrier on the surface of a thin film nanofibrous composite (TFNC) membrane. Self-assembly of GO sheets with nanoscale thickness led to the formation of the barrier layer. The low TFNC transfer barrier mat provided a unique advantage because of its high bulk porosity (80%) with pore structures that were fully interconnected, leading to extraordinary improvement in separation performance over pristine PVA membranes, as shown in Figure 17. The authors claimed that the performance of GO-based TFNC membranes may be improved by barrier layer thickness optimization.

Graphene oxide (GO) composite membranes have also been immobilized on polyacrylonitrile (PAN) substrates through the incorporation of copper hydroxide nanostrands (CHNs) between GO sheets, coupled with an extremely thin layer of hydrophilic sodium alginate (SA), as described by Hao et al. [78]. The authors claimed that the homogeneous intercalation of CHNs in the GO membrane enhanced the channels of water permeation without disrupting the GO interlayered structure. The top layer of SA attached onto the CHNs-containing GO membrane through electrostatic interaction and hydrogen bonding, preventing the swelling of the GO membrane and simultaneously enriching the water molecules. This served as a promising application in pervaporation for the dehydration of ethanol with a long-term stability. Liang et al. [79] reported the application of a facile and tunable to incorporate a type of multifunctional polyhedral oligomeric silsesquioxane (POSS) into the GO interlayer channels. They showed that GO channels with a high separation accuracy and sturdy framework can be developed from the hydrophilic-hydrophobic hybrid structure, along with the covalent cross-linking sites on POSS, respectively. The hydrophilic–hydrophobic hybrid structure facilitated the preferential adsorption and the quick diffusion of water molecules at the same time; and the abundant covalent cross-linking sites led to strong bonding between POSS and GO. The fabricated membranes achieved a significantly high flux of 3.16 kg/m^2^ h and a separation factor of 1303 for water/ethanol separation, which are 50% and 31-fold higher as compared to the pristine GO membrane. Tang et al. [80] demonstrated a novel modification method of graphene oxide (GO) by the addition of an ionic liquid (IL) and incorporated the modified GO into a polyether block amide (PEBA) membrane to separate butanol aqueous solutions via pervaporation. The ionic liquid has strong affinity towards butanol with a hydrophobic characteristic, the adsorption selectivity of butanol over water by the IL-modified GO was 4 times greater than that of pristine GO. The results of pervaporation showed that the addition of IL-GO enhanced the mixed matrix membrane performance by improving the separation factor and permeation flux by by 31.5% and 18.2%, respectively (by 1 wt% content of IL-GO), as compared to pristine PEBA membrane.

### 2.5. Cellulose Nanocrystals

Cellulose nanocrystals (CNCs) are emerging as an eligible component for membrane-based applications because of their high specific surface area and tunable surface properties. CNCs are basically the crystalline regions that have been extracted from cellulose microcrystals by the hydrolysis of strong acids at extremely high temperatures. CNC appears as elongated rod-like structures that are crystalline in nature with negligible flexibility because of the absence of amorphous regions. However, the presence of relatively polar surfaces in CNCs enhance the passage of polar species such as water through non-polar matrices, by building transport channels that are polar in nature at the nanocellulose-nonpolar matrix interface [81,82,83]. Bai et al. [84] developed pervaporation membranes from poly(vinyl alcohol) (PVA) with varying amounts of cellulose nanocrystals as filler for separating ethanol-water mixtures which was used as a model system. The PVA/cellulose nanocomposite membrane composed of 1 wt% cellulose nanocrystals exhibited the best performance in pervaporation, whose average permeate flux slightly reduced but an increase in separation factor from 83 to 163 was observed for an aqueous solution with 80% ethanol at 80 ℃ respectively. A very recent article led by Kamtsikakis et al. [85] reported nanocomposite membranes based on a hydrophobic poly(styrene)-block-poly(butadiene)-block-poly(styrene) (SBS) matrix and CNCs that exhibit water transport with directional properties for the recovery of ethanol in an ethanol–water model system. The CNCs were further modified with hydrophobic oleic acid moieties (OLA-CNCs) to understand the effect of this modification on the morphology and separation performance of SBS/OLA-CNC nanocomposite membranes. It is noteworthy that SBS/CNC membranes exhibited either improved or decreased mass fluxes as compared to the base SBS membranes, but also a steady reduction in membrane selectivity towards ethanol, as shown in Figure 18. The authors concluded that by tuning the surface chemistry of CNCs, the pervaporation performance of the fabricated membranes and the level of asymmetry in mass transport can be tuned.

### 2.6. MXenes

Mxene is a new class of 2D materials and has a formula of *M_n+1_X_n_T_x_*, where *n* is 1, 2, or 3, *M* is an early transition metal, *X* represents C and/or N, *T* is the surface group (OH, O, or F). Mxene has been used for the synthesis of separation membranes for water desalination [86], ion sieving [87], and gas separation [88]. The hydrophilic nature and laminated structure of these MXene based membranes showed quicker and selective permeation of water molecules. A recent study led by Xu et al. [89] showed the potential applications of MXene-based membranes in the dehydration of organic solvents, e.g., water/ethyl acetate, water/ethanol, or water/dimethyl carbonate mixtures, by pervaporation. Ti_3_C_2_T_x_ were exfoliated into the chitosan matrix and delaminated into Ti_3_C_2_T_x_ powders. The authors concluded that the surface sorption remained unchanged after the incorporation of MXene nanosheets, while the water permeation through the membrane was greatly enhanced due to interlayer channels of the assembled MXene laminates, thereby improving both permeate flux and separation factor. Particularly, the optimized 3 wt% MXene/CS MMM exhibited a total flux of ~1.4–1.5 kg/(m^2^ h) and separation factor of 1421 for the ethanol dehydration at 50 °C, as shown in Figure 19.

Wu et al. [90] demonstrated the role of pristine MXene membrane for alcohol dehydration using a MXene membrane with a thickness around 2 μm, coated with a monolayer of Ti_3_C_2_T_x_ nanosheets for the separation of ethanol-water mixture via pervaporation. The water/ethanol total flux and separation factor obtained by the MXene membrane was 263.4 g m^−2^ h^−1^ and 135.2 respectively, using an ethanol concentration of 95% at room temperature. Thus, MXenes exhibit a promising future in the separation industry. The authors noted that the dehydration performance of the MXene membrane is not significantly high compared to the commercial zeolite membranes, such as NaA due to much longer mass transfer path giving lower flux, higher swelling and defects in the layered structure which reduces separation factor. Some ways to improve the performance of MXene membranes include lowering the thickness of the membrane, chemically modifying the membrane surface to improve solubility, and better designing the 2D nanochannels to enhance diffusivity. It is also believed that MXene membranes that are stacked with smaller radial sized nanosheets will also exhibit greater permeance and create some nanopores or sub-nanopores on the surface of the nanosheets that will lead to enhanced transport. In a recent article [91], a new membrane fabrication strategy for solvent dehydration is proposed in which macromolecules are present in between the MXene nanosheets after which interfacial polymerization takes place on the arranged laminar membrane surface (Figure 20). The ordered stacked structures were built by the electrostatic interaction between the negatively charged MXene nanosheets and the positively charged hyperbranched polyethylenimine (HPEI). At the same time, in order to close possible defects that are non-selective in nature, an interfacial polymerization reaction took place between trimesoyl chloride (TMC) and HPEI. This is the first article where for molecular separation Ti_2_CT_x_ nanosheets were developed as a new class of 2D-material membranes because of their higher hydrophilic nature as compared to that of Ti_3_C2T_x_, for use in the solvent dehydration. An excellent performance of water/isopropanol mixture dehydration was observed for the defect-free Ti_2_CT_x_-based membranes (thickness around 100 nm), with permeate water concentration of more than 99 wt% and better performance than Ti_3_C2T_x_ due to sorption and size sieving effects.

Cai et al. [92] prepared mixed matrix membranes (MMMs) by mixing Ti_3_C2T_x_ with the PVA matrix, and was tested on an ethanol-water binary system via pervaporation. The PVA/Ti_3_C2T_x_ MMMs were compatible and exhibited resistance to swelling. The separation factor of the MMM was significantly enhanced because the membrane cross linking density increased due to the presence of Ti_3_C2T_x_. The optimum separation performance was achieved with 3 wt.% Ti_3_C2T_x_ loading, acquiring a separation factor of 2585 and a cumulative flux of 0.074 kg/m^2^ h for the separation of a 93 wt% ethanol at 37 °C.

Li et al. [93] showed that single or double layered MXene nanosheets with two-dimensional interlayer gaps and optimum hydrophilicity are promising candidates as porous fillers for the developing MMM with a high performance capability for pervaporation dehydration of ethanol/water mixtures. Single or double-layered MXene nanosheets need exfoliation from multi-layered MXene using ultrasonication for longer periods of time and then centrifugation at high speed, which makes large-scale application a problem. Li et al. fabricated sodium alginate (SA) MMM by preparing intergap mMXene particles under mild conditions. Compared with the pure SA membrane, the SA/mMXene MMM had increased hydrophilicity but with reduced swelling at higher mMXene content. Moreover, the MMM membrane having 0.12 wt% mMXene in SA shows 10× separation factor (9946) and the permeate flux was 24.5% lower (505 g m^−2^ h^−1^) for ethanol/water (90 wt%) solution dehydration, owing to preferential water sorption and restricted ethanol diffusion.

By-products of the paper and agricultural industries include lignosulfonates. In a study led by Li et al. [94], a pervaporation membrane was developed by mixing hydrophilic calcium lignosulfonate (CaLS) with single or double-layered MXene on sodium alginate (SA) to prepare for ethanol dehydration (Figure 21). CaLS not only enhanced the hydrophilicity, but also decreased membrane swelling, and MXene led to the development of a layered cross-sectional structure that further decreased swelling. From the results, it was evident that membrane permeation flux and separation factor enhanced by 74% and 160%, respectively, at 90 wt% ethanol in the feed. MXene also improved the pervaporation performance of the hybrid membrane, with the permeation flux and separation factor of approximately 938 g·m^−2^·h^−1^ and 4612, respectively, thus opening a pathway for theoretical and technical studies to expand the efficient usage of lignosulfonates.

### 2.7. Covalent Organic Frameworks (COFs)

The production of porous, crystalline, and stable materials is facilitated by the formation of covalent organic frameworks (COFs), two- or three-dimensional structures that originate from chemical interactions between organic precursors. Because the researchers were able to pick the best precursors and manage the synthesis process, COFs are widely known. Converting non-porous and amorphous organic compounds into porous and crystalline ones that offer remarkable material stability in a variety of solvents and environments was greatly aided by these advancements in coordination chemistry. Covalent organic frameworks (COFs) have recently discovered several uses in gas separation [95,96], membrane pervaporation [97], and filtration processes, e.g., ultrafiltration [98,99].

Wu et al. [100] employed COF-LZU1 particles that were made in a poly(ether-block-amide) (PEBA) matrix by mixing benzene-1,3,5-tricarbaldehyde (TFP) solution into the PEBA casting solution, drying for membrane development, and then utilizing p-phenylenediamine (PDA) solution on the PEBA membrane’s surface. According to this publication, PDA molecules were able to pass through the membrane and interact with TFP molecules to produce COF-LZU1 in-situ. The PEBA matrix’s constricted structure substantially hindered COF-development LZU1’s and aggregation, which resulted in COF-LZU1 that was evenly distributed throughout membranes. With a separation factor of 22.2, which was 139% greater than pristine membranes, and a permeation flow of 611 g/m^2^h, the mixed matrix membranes showed enormous promise for the recovery of n-butanol. Wu et al. [101] also fabricated two dimensional COF-LZU1 with p-phenylenediamine and 1,3,5-triformylbenzene that was shown to have excellent stability, hydrophobicity, and wide pores (1.8 nm) that promoted alcohol sorption and diffusion for diluted n-butanol solutions. The hydrophobicity and n-butanol partition coefficient rose with larger COF-LZU1 concentrations, whereas the flow and separation factor increased before decreasing, demonstrating anti-trade-off effects. The authors demonstrated that by reducing the selective layer’s thickness and raising the feed solution’s temperature, membrane performance may be improved. The permeation flux and separation factor of MMMs (thickness of 21 μm and 64 °C) were 2694 g m^−2^h^−1^ and 38.7, respectively, as shown in Figure 22.

The post-synthetic linker exchange (PLE) method was explored recently to fabricate COF membranes for dehydration of alcohols as shown in Figure 23 [102]. In order to improve the material’s molecular sieving properties, the PLE approach utilized the reversible breaking-reformation process between unmodified COF membranes and the linkers that were employed. The procedure increased the hydrophilic groups on the COF membranes’ surface, which enhanced water sorption. To reduce flaws and improve the COF membranes’ sieving abilities, the gap between neighboring COF crystals was eliminated. For effective butanol dehydration, these membranes’ surface microenvironment and pore and channel architecture were improved by the generated Hz monomers. The developed COF membranes showed high n-butanol dehydration performances with a selectivity of 3620, 11.35 kg m^−2^ h^−1^ of total flux, and were found to be stable during extended operation.

Li et al. [103] suggested organophilic porous particle doping that has H-bonding interaction sites within pore channels to enhance performance of PDMS membranes in pervaporation as shown in Figure 24. COF-300 was fabricated and mixed with PDMS to form mixed matrix membranes (MMMs) to demonstrate superior affinity toward furfural with an adsorption value of 525.3 mg g^−1^ at high temperatures (80 °C). PDMS membranes’ ability to pass through furfural, aniline, butanol, ethanol, and phenol with the least amount of water transport was demonstrated through pervaporation studies. Water’s permeability was decreased as a result of the H-bonding between its molecules and COF-300, which also increased the mass transfer resistance of the water molecules. Due to hydrogen bonding, COF-300’s higher affinity for organic moieties makes it a viable material to facilitate the transport of organics while preventing the diffusion of water. Furfural permeability was improved by 14.1% when compared to unmodified PDMS membranes, whereas water permeability dropped by 20.0% when furfural was separated from an aqueous solution at 80 °C. The potential of the recommended method and COF-300 in creating membranes for the separation of organic from aqueous solutions was shown by the improvement in selectivity of 42.7%.

COF-based membranes are still under development with researchers focusing on new hierarchical structures. Yang et al. [104] fabricated a novel approach for developing hollow COF nanospheres. Due to the hydrophilic groups and increased stability, an imine-linked COF TpBD made from 1,3,5-triformylphloroglucinol (Tp) and benzidine (BD) was chosen as the core component. In order to create water-selective membranes for ethanol/water separation, sodium alginate (SA) matrices were then filled with the manufactured TpBD (H-TpBD) nanospheres. The nanospheres produced excellent diffusion tracks and a large number of water-interaction sites, which facilitated rapid water penetration across the membranes. The SA membrane’s hydrophilicity was increased by the inclusion of -NH-/-NH2 groups from H-TpBD, increasing the solubility selectivity for water molecules. Due to the hydrogen bonding affinity between SA and H-TpBD and the organic makeup of H-TpBD, sized adjusted free volume cavities were produced with a separation factor of 2099 and a permeation flow of 2170 g/m^2^h.

A type of COF membrane was proposed by Yang et al. [105] employing 1D cellulose nanofibers (CNFs) and 2D COF nanosheets as the construction pieces. Due to their hydrogen-bonded parallel chains, 1D CNFs are among the strongest natural materials, and the hydroxyl groups on their surface can be exploited for surface functionalization. In a single process, the tightly interlocked COF membranes are constructed from the mixed-dimensional COF nanosheets covered by CNFs. A strong interlamellar microporous network is produced by the sheltering action of CNFs, which also reduces the size of the pore entrance of COF nanosheets. For the dehydration of n-butanol, the manufactured membranes had a flow of 8.53 kg/m^2^h and a separation factor of 3876.

Wang et al. [106] used a Brønsted acid mediated one-step self-assembly method for the fabrication of COF membranes by segregating the organic phase (containing Brønsted acids and aldehydes) from the aqueous phase (containing amines) with a polymeric support and implementing an interfacial polymerization reaction. Figure 25a,b depict the COF-Schiff JLU2’s base reaction on the support. Brønsted acids play a crucial role in controlling the microstructure evolution of COF-JLU2 membranes by facilitating the amorphous-to-crystalline transition, ensuring the restricted membrane growth at the interface, and regulating the assembly behavior of COF subunits. The n-octanoic acid-mediated membrane showed a better separation factor of 5534 and a total flow of more than 10,573 g/m^2^h^−1^ for butanol dehydration.

Fan et al. [107] demonstrated the recovery of n-butanol from aqueous solution using composite materials of supported COF-based membranes made by inserting hydrazone-linked COF-42 into commercial hydroxyl-polydimethylsiloxane (PDMS). For the separation of an aqueous solution containing 5.0 weight percent n-butanol at 80 °C, the COF-42-PDMS membrane demonstrated a high separation factor of 119.7 with a total flux of 3306.7 g/m^2^h. Approximately 86.3 weight percent n-butanol was recovered in the permeate side. The amphipathic COF-42 present in the membrane selective layer, which simultaneously improves the adsorption of n-butanol and water molecules, is the primary cause of the increase in permeation flow. The strong selectivity can be attributed to n-butanol molecules diffusing more quickly through the COF-based membrane than water molecules do. In this work, the COF-42-PDMS membrane was effectively coupled to create a true non-distillation process that produced fuel-grade biobutanol.

Covalent organic nanosheets (CONs) and poly(ether sulfone) were used to create hybrid membranes with hierarchical pore architectures that imitate the respiratory system in living things [108]. By using the Schiff base reactions of 1,3,5-triformylphloroglucinol (Tp) with p-phenylenediamine (Pa-1) and benzidine (BD), two types of amide modified CONs, TpPa-1 and TpBD, with varied pore diameters were created and individually integrated into the polyether sulfone matrix (PES), as shown in Figure 26. Remarkably, the hybrid membrane composed of TpPa-1-CON (8 wt%) exhibited a high water/ethanol separation factor of 1150, while the membrane composed of TpBD-CON (8 wt%) exhibited a high water/ethanol separation factor of 1150. The separation factor for /n-butanol was 2735. The permeation fluxes for both separation systems were higher than 2.5 kg m^−2^ h^−1^. The researchers concluded that the relationships between hierarchical structures and membrane performance might make it easier to develop innovative membrane architectures that have superior features.

### 2.8. Transition Metal Dichalcogenides (TMD)

Molybdenum disulfide (MoS2), a transition metal dichalcogenide that resembles graphene on a two-dimensional scale, has three crystal phase states: 1 T-MoS_2_, 2H-MoS_2_, and 3R-MoS_2_. Superior elasticity, flexibility, and excellent mechanical strength are all features of MOS_2_ [109,110]. Rajan et al. reported five times faster water transport on the surface of the MoS_2_ nanosheet than that on the surface of the GO nanosheet [111]. In addition, the monatomic thick MoS_2_ nanosheets produce smooth and fixed nanochannels due to the absence of functional groups [110,112]. This opened the door for research into MoS2 functional layers with nanochannels that would speed up water diffusion by giving water molecules a shorter path to travel and molecular sieve nanochannels of stacked MoS_2_ nanosheets that would enhance the hybrid membrane’s separation performance [113]. The creation of a hybrid membrane using MoS_2_ material for pervaporative alcohol dehydration was only recently performed by Taymazov et al. [114]. In order to minimize the macropores on the ceramic hollow fiber (CHF) membrane’s surface and create the TMD layer, an intermediary layer of titanium dioxide (TiO_2_) was built. Polyethyleneimine (PEI) was utilized as a binder as shown in Figure 27. In an aqueous solution of 90 weight% isopropanol at 343 K, the manufactured MoS_2_ hybrid membrane had a permeation flux of 5697 g/m^2^h and a separation factor of 320.

In a recent publication, sodium alginate (SA) was used with MoS_2_ nanosheets to separate ethanol from water [115]. When compared to SA pure membrane, the microstructure of the membranes was changed by the addition of MoS_2_ nanosheets, and the swelling degree was decreased by 33%. According to the authors, quicker transport routes for water penetration were made possible by the ordered stacking of MoS_2_ nanosheets and their interlayer spacing. The permeation flow was 1839 g/m^2^h and separation factor was 1229 in ethanol/water (90/10 wt%) solution at 350 K, which were 54% and 85% higher than those of SA/PAN pure membrane, respectively.

### 2.9. Metal Organic Framework (MOFs)

A family of substances known as metal-organic frameworks (MOFs) is made up of metal ions or clusters that are coordinated to organic ligands to create one-, two-, or three-dimensional structures. They belong to the category of crystalline materials that are porous and contain both organic and inorganic elements. Metal-organic frameworks (MOFs) are thought to be effective adsorbent materials because of their high specific surface areas and crystalline structure [116,117]. MOFs have been used in CO_2_ capture and hydrocarbon separation [118,119,120]. Recently, research has also been conducted on MOFs’ possible use in water–ethanol separations. The hydrophilic MOF-801 crystals were anchored into the chitosan (CS) matrix to create MOF-801/CS mixed matrix membranes (MMMs) for pervaporation dehydration of ethanol. These MMMs allow for the selective penetration of water through the porous filler [121]. Using experimental tests and modeling, it was shown that MOF-801 crystals selectively adsorb water molecules and that low energy sorption sites enable ethanol molecules to diffuse through MOF-801 with little energy. Enhanced flux and separation factor are produced by the porous structure of MOF-801, which offers more transport paths for water molecules and makes ethanol molecules’ transport pathways more convoluted. The manufactured membrane had a separation factor of 2156 and a total flow of 1937 g/m^2^h when MOF-801 was loaded at 4.8 weight% of the membrane.

Due to the interaction between the surface SiOH and the ethanol, MFI zeolite membranes made in an alkaline environment are unstable when used to separate an ethanol/water combination. The ensuing oxy-ethyl groups prevented the zeolite channel from entering and reduced the size of the effective orifice, which caused the flow and separation factor to drop and called for modifying the MFI membranes. As shown by Wu et al. [122], dopamine modification of MFI zeolite membrane can successfully reduce silanol impact and enhance long-term separation stability. However, the primary methods for water–ethanol separation with well-designed MOFs depend on the variations in the forces of interaction between MOFs and either water or ethanol, the pressure-dependent gate-opening effect of MOFs, and the space limitation effect caused by the molecular size and fixed channel structure. The selectivity of separations based on contact force differences is substantially lower. Wang et al. [123] To separate mixtures of ethanol and water, a sturdy Cu-triazole metal-organic framework (CuTria) with linked 3D supermicroporous channels, tridentate bridging triazole ligands, and abundant open Cu^2+^ sites was developed. This MOF demonstrated an optimal molecular sieving effect for ethanol/water separation due to the size exclusion effect for ethanol and good water affinity. The findings of a dynamic breakthrough experiment with water/ethanol (*v/v* = 5:95) mixtures showed that this MOF can successfully separate ethanol and water in its whole and produce pure ethanol vapors with a purity level better than 99.99%.

Jiang et al. [124] modified ligands with different functional groups for pervaporation dehydration to examine the impacts of MOFs. To create a water-selective membrane for ethanol dehydration, three hydrophilic UiO-66-X nanoparticles were created and combined with a poly (vinyl alcohol) matrix. According to the scientists, UiO-66-X’s counterbalancing features, improved polarity, and a sufficient pore size led to the material performing at its best when it came to dehydrating ethanol/water mixtures. Figure 28 illustrates the UiO-66-X nanoparticles’ pore size distribution data at 5–8 and 12–16 Å, which is larger than the kinetic diameters of water (2.7 Å) and ethanol (4.2 Å). Due to the rapidly rising weight of the framework and, more critically, the decreased crystallinity, the surface areas and pore size of UiO-66-X nanoparticles decrease with increasing ligand polarity. Water molecules might be activated due to the UiO-66-(COOH)_2_ nanoparticles’ strong contact with water molecules and the continuous pathways for water diffusion that were given by their pore aperture. This interaction helped the water flux by enhancing the penetration and diffusion of water molecules. The hybrid membranes with UiO-66-(COOH)_2_ loading of 8 wt% exhibited a total flux of 979 ± 7 g/(m^2^·h) and a separation factor of 2084 ± 21.

## 3. Limitations and Future Perspectives

When compared to distillation and other non-membrane-based technologies, these nanomaterial-based membrane separation methods for the recovery or dehydration of organic solvents have achieved significant strides in terms of energy efficiency and use as biofuels. For technology to be commercialized, however, several obstacles still need to be overcome, including shorter lifetime, a poor separation factor, membrane fouling, and the expensive cost of membranes. New membranes with better separation and flux capabilities are needed for commercial use in the use of membranes to fermentative separations. The material reviewed in this article makes apparent that MMMs are now some of the highest-performing membranes for these separations, yet a number of issues have been identified with MMM manufacturing.

Although efforts have been made to take use of MOF-based membrane design methodologies, various limitations of these methods may limit their practical application. The micropores of MOFs may collapse following hybrid procedures for those MOFs with reducing functional groups and polyvalent metal nodes as a result of the oxidation of the functional groups or metal nodes. Additionally, the maximum loading of metal ions in the self-assembly method limits the number of MOFs in the MOF-based composite membrane. The ultimate loading quantity is determined by the density of functional groups on the membrane’s surface and the strength of the bonds that connect groups and precursors. Increased surface modification technologies and logical functional group selections may successfully raise the loading level. However, a large loading quantity may cause MOFs to aggregate, which reduces the ability of MOF-based membranes to conduct separation. Therefore, one of the key problems in the design of MOF-based membranes is to find the essential balance between the loading quantity, the dispersibility of MOFs, and the separation performance of the produced membrane.

Future industrial production applications of MOF-based membranes still face difficulties. On the one hand, there is a dearth of experience describing their long-term stability under hypothetical separation scenarios. It is necessary to conduct more research on the long-term durability of MOF-based membranes in acidic/basic environments, complex organic solvent systems, and high temperatures. However, the comparatively expensive cost of MOFs may make it difficult to use MOF-based membranes widely. For the purpose of boosting the industrial uses of MOF-based membranes, more cost-effective design techniques should be created. It should never be given up trying to find well-defined MOF-based membranes that nonetheless have good liquid separation capabilities. Future work should concentrate on creating MOF-based membranes with multifunctional separation capabilities in additional environmental sectors, in addition to long-term stability under realistic liquid separation circumstances.

The adjustable physicochemical features, in-plane pore structure, and inter-layer 2D channels of Mxene, which are ascribed to its apparent potential as an emerging material for next-generation separation technologies, were emphasized in this review. By reducing these technical barriers, it is anticipated that more and more MXene separation applications will be investigated in the future, utilizing the full potential of this wonder substance and replacing many other materials. Continuous effort, devotion, and diversified research endeavors might make this achievable. The comments above make it quite evident that there are several problems and difficulties that require attention. The manufacture of MXene, which is exceedingly costly and causes a delay in its widespread usage, comes first. More techniques are being developed to create defect-free, large-area MXene with controlled pore size, shape, and interlayer spacing. Recently developed technologies have played a crucial role in reducing the cost of material to some extent. In the future, it will also be necessary to design a suitable substrate to prevent dispersion in the liquid phase and to guarantee water conveyance.

Developing graphene-based membranes presents several challenges, including how to create holes for high selectivity. Although this approach has been the most frequently used, it is not always simple to implement, especially when the pore size needs to be controlled to sub-nanometers. The most common method has been to control the size of the nanopores in the membranes to be comparable to the size of molecules to be separated. Future research should aim at the possibility of achieving high selectivity without the use of nanopores with sub-nanometer diameters by utilizing the ionization of the numerous functional groups that are present in graphene-based membranes. More research has been conducted on graphene-based membranes for gas separation and water treatment than on those for the pervaporation separation of organic solvents. More theoretical and experimental research is required to gain a deeper understanding of the molecular transport within the graphene-based membranes based on the mechanisms of pervaporation. The targeted combinations should guide how the graphene-based membrane structures are modified.

It is crucial to evenly distribute the fillers throughout the mixed matrix structure. Since the inter-filler free channel space was often too broad to be molecularly selective, in the event that fillers aggregated, the “channel flow” would dominate the mass transfer throughout the agglomerate’s region. Consequently, depending on the percentage of the fillers generated in the agglomerates, the highly discriminative flow that occurs in the interior or over the smooth surface of the CNTs would lose its opportunity to improve membrane selectivity. It should be mentioned that in order to ensure the highly selective performance, the dispersed fillers also must have strong interfacial compatibility with the polymer matrix to prevent the non-selective “leaky flow”. More research should be performed to determine how various methods for fabricating membranes affect how well MMMs function for relevant separations in fermentations. Further testing of several water stable MOFs for use as inorganic fillers should also be performed. Additional research on fouling and long-term stability of high-performing MMMs will aid in the development of strategies to combat fouling caused by the many different components of microbial fermentation broths.

## 4. Conclusions

One of the most energy-intensive stages in the generation of renewable energy is the separation of organic solvents, which raises the cost of the entire process. As a result, choosing effective separation technologies is necessary to commercialize sustainable energy generation. Many alternatives to traditional distillation have been put out in recent times. This study discusses current advancements in membrane-based separation employing innovative materials for organic solvent separation. At this point, it is crucial to construct membranes for separation based on 2D nanomaterials. Solid–liquid or solid–gas interactions provide separation through the presence of interlayer channels and nanopores. A trade-off between selectivity and permeability has always existed, giving birth to surface functionalization and better interface conditions. The majority of current papers on membranes based on nanomaterials are experiment-driven. Future views could include simulation/modeling techniques to comprehend the design of membranes based on nanomaterials. The performance and characteristics of membranes should be enhanced in the meantime. It is crucial to keep researching innovative mixed matrix membranes with enhanced resilience in harsh settings. There is considerable space for enhancing the separation capabilities of these composite membranes by careful modification of the MOF/polymer interface morphology. However, using nanomaterials will result in increased expenses. To properly examine the payback period and total cost benefits of employing nanomaterials in membrane fabrication, thorough economic analysis must be performed. The creation and use of 2D nanomaterial-based membranes for the separation of organic solvents are both areas that the authors of this paper think will garner increased interest.

## Figures and Tables

**Figure 1 membranes-13-00108-f001:**
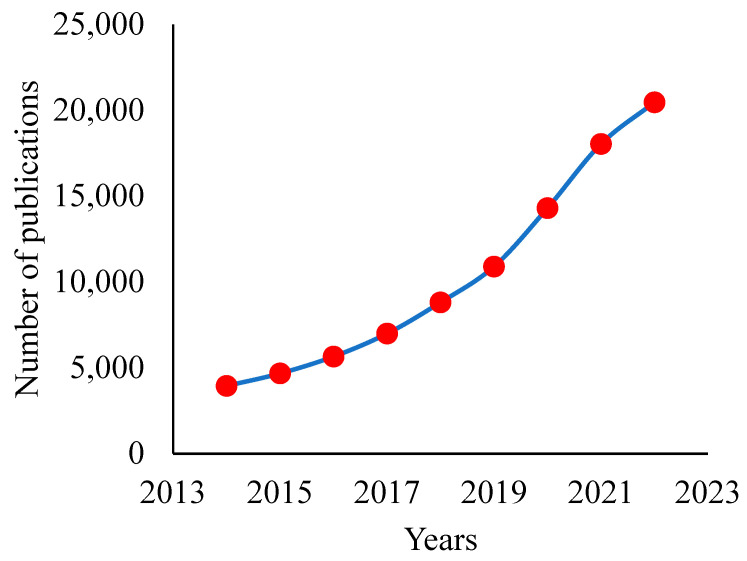
Publication trend of articles on fabricated membranes for organic solvent separation.

**Figure 2 membranes-13-00108-f002:**
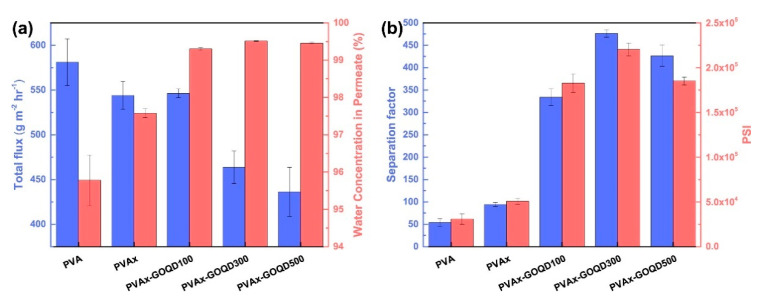
Dehydration performance of pervaporation membranes in the separation of 70/30 wt% i-PrOH/water mixture at 25 °C: (**a**) Water concentration and cumulative flux in permeate and (**b**) pervaporation separation index (PSI) and separation factor.

**Figure 3 membranes-13-00108-f003:**
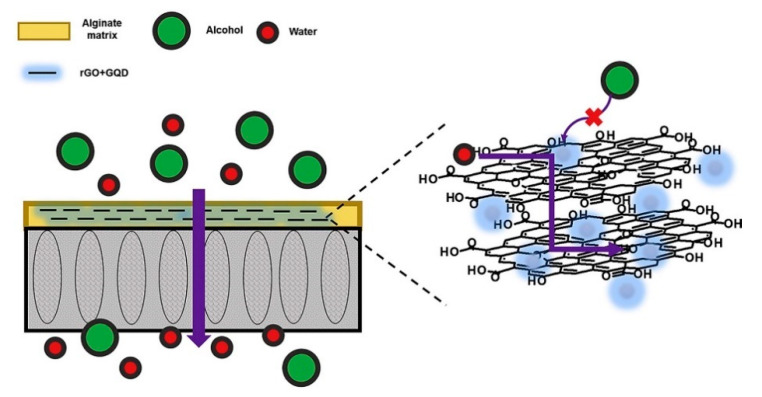
Proposed mechanism of sealing structural defects using Graphene Quantum Dots (GQD).

**Figure 4 membranes-13-00108-f004:**
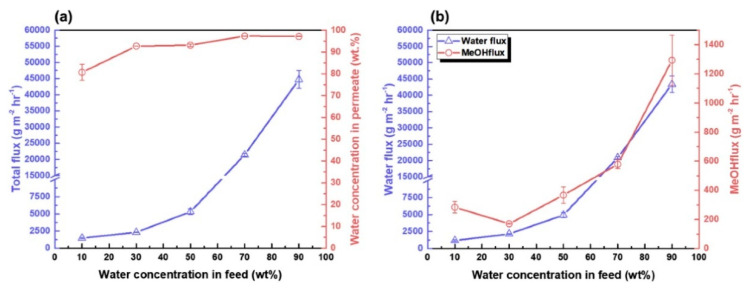
Effect of feed temperature on (**a**) cumulative flux and permeate water concentration in the separation of a 70/30 wt% MeOH/water using an Alg_rGO + GQD_3 membrane, (**b**) water and MeOH individual flux.

**Figure 5 membranes-13-00108-f005:**
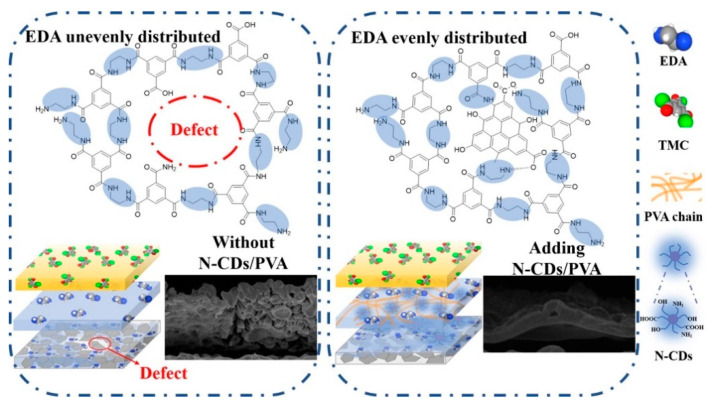
Schematic representation of the IP reaction mechanism: with no N-CDs and PVA (**Left**); including N-CDs and PVA (**Right**).

**Figure 6 membranes-13-00108-f006:**
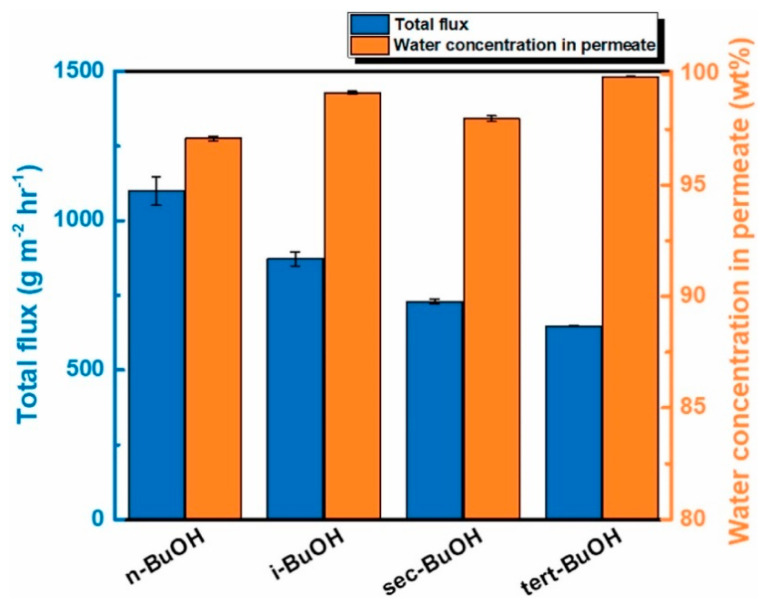
Performance of the pervaporation procedure for separating various butanol isomer/water combinations at 25 °C using the TA0.075_TMC0.4_NGQD(50) membrane.

**Figure 7 membranes-13-00108-f007:**
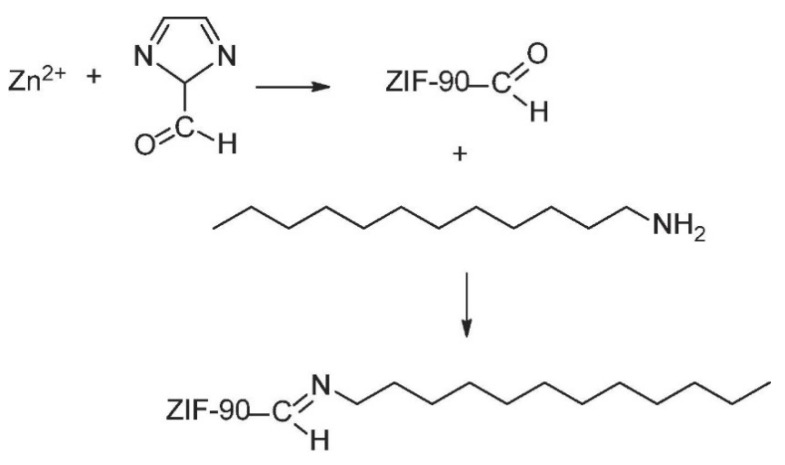
Schematic of the chemical reaction and DLA modification of ZIF-90 particles.

**Figure 8 membranes-13-00108-f008:**
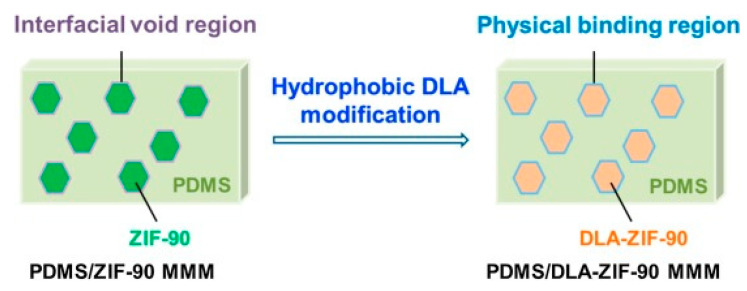
Schematic of physical binding between DLA-ZIF-90 particles and the PDMS matrix.

**Figure 9 membranes-13-00108-f009:**
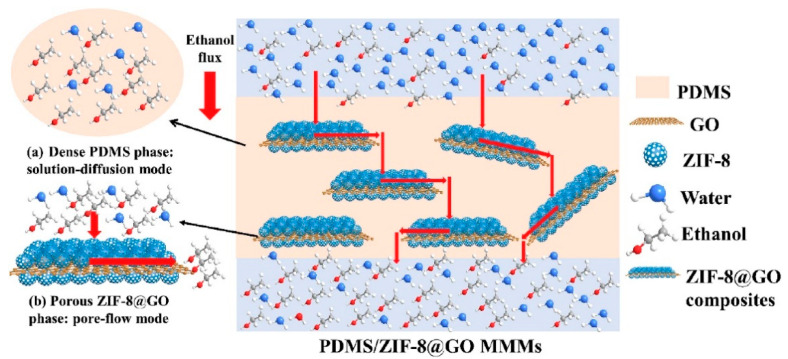
Schematic representation of mechanism of PDMS/ZIF-8@GO MMMs in ethanol recovery.

**Figure 10 membranes-13-00108-f010:**
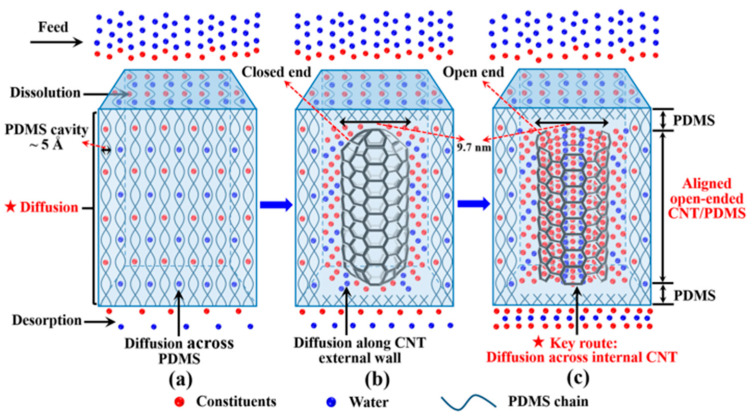
Schematic of the mechanism of molecule transport (**a**) The pure PDMS membrane; (**b**) the aligned CNT (closed-ended)/PDMS membrane; (**c**) the aligned CNT (open-ended)/PDMS membrane.

**Figure 11 membranes-13-00108-f011:**
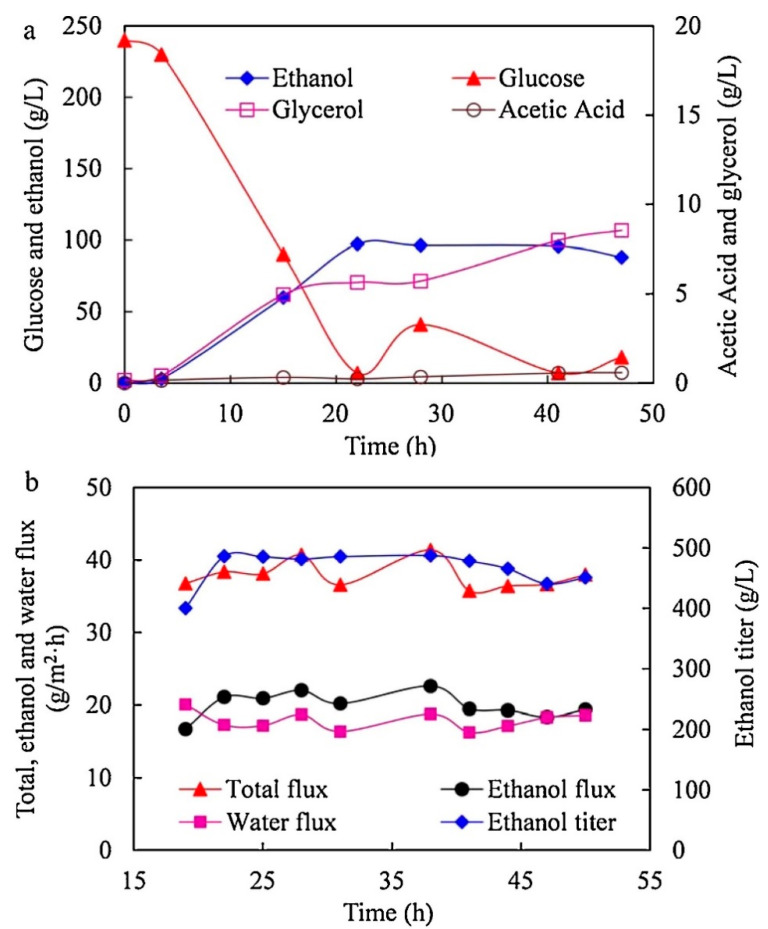
Ethanol fermentation based on pervaporation with a CNT-mixed PDMS membrane. (**a**) the rate at which the concentration of glucose and other products in the fermentation broth changes; (**b**) the efficiency of the pervaporation membrane during the fermentation of ethanol.

**Figure 12 membranes-13-00108-f012:**
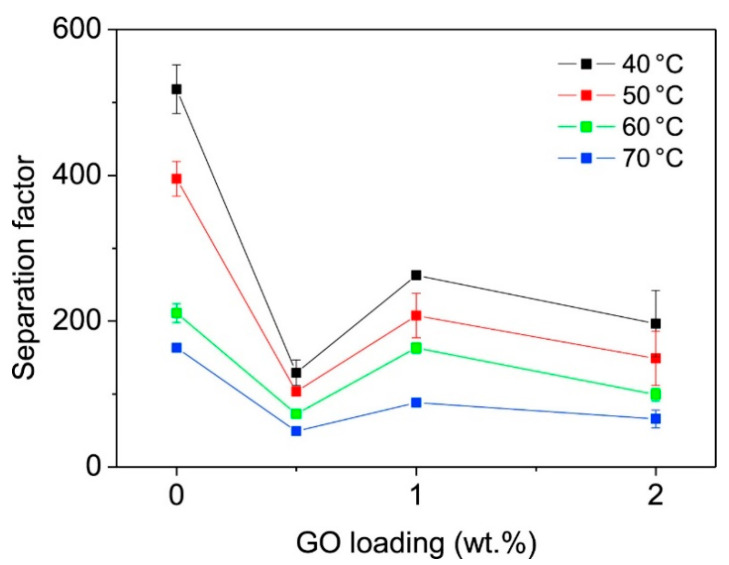
Cumulative permeate flux for a 10:90 wt% water-ethanol combination under various GO loading scenarios and at various operation temperatures.

**Figure 13 membranes-13-00108-f013:**
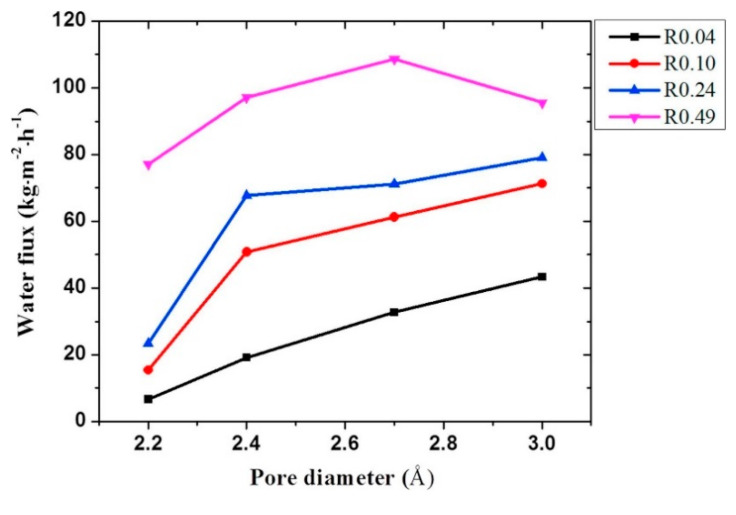
Water flux across GO monolayer membranes based on variable pore diameter and O/C ratio (R).

**Figure 14 membranes-13-00108-f014:**
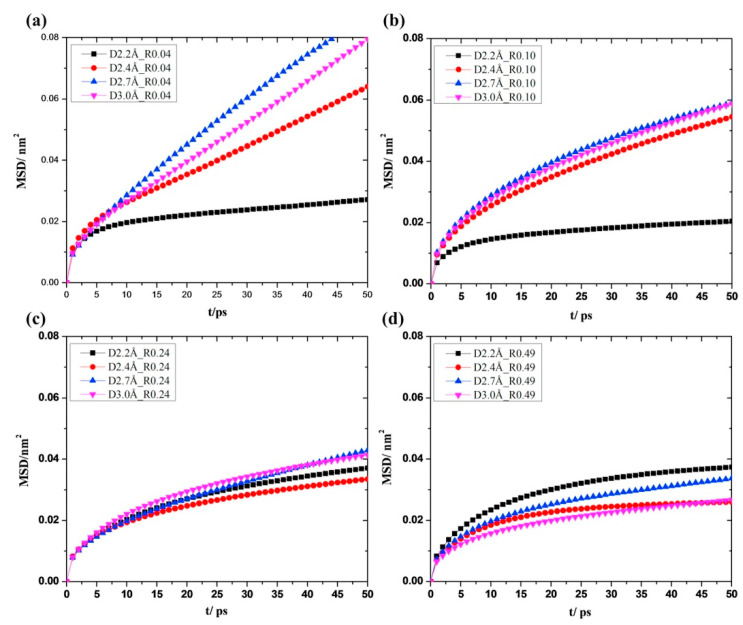
Based on varying GO membrane oxidation levels, water diffuses down the z-axis at rates of (**a**) R = 0.04; (**b**) R = 0.10; (**c**) R = 0.24; and (**d**) R = 0.49. The relative impact of pore diameter has been depicted in each illustration using a variety of colors.

**Figure 15 membranes-13-00108-f015:**
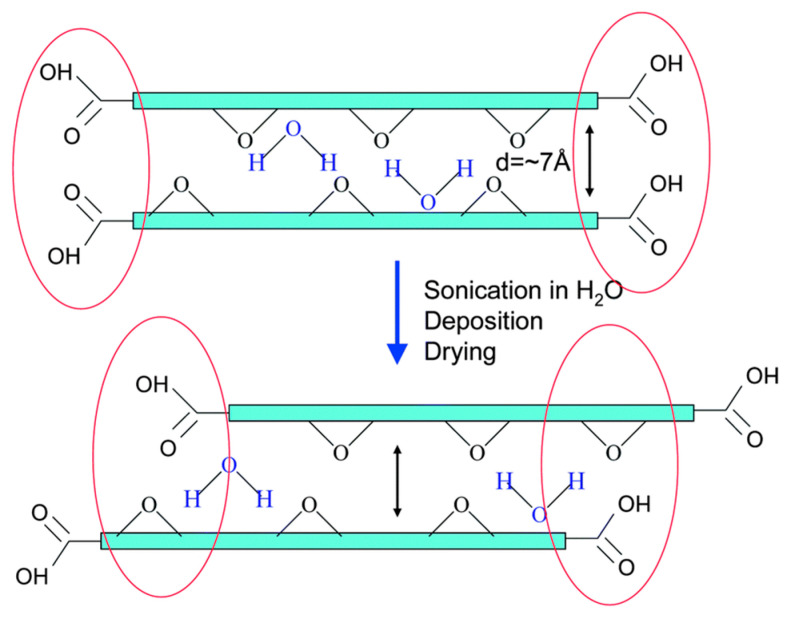
Schematic showing the arrangement of graphene oxide flakes in powder and membranes. Various combinations of functional groups govern the access to the interlayers.

**Figure 16 membranes-13-00108-f016:**
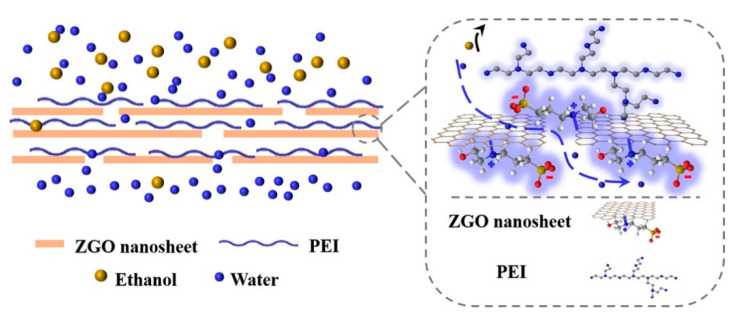
Schematic representation of the water channels present in ZGO-based membranes.

**Figure 17 membranes-13-00108-f017:**
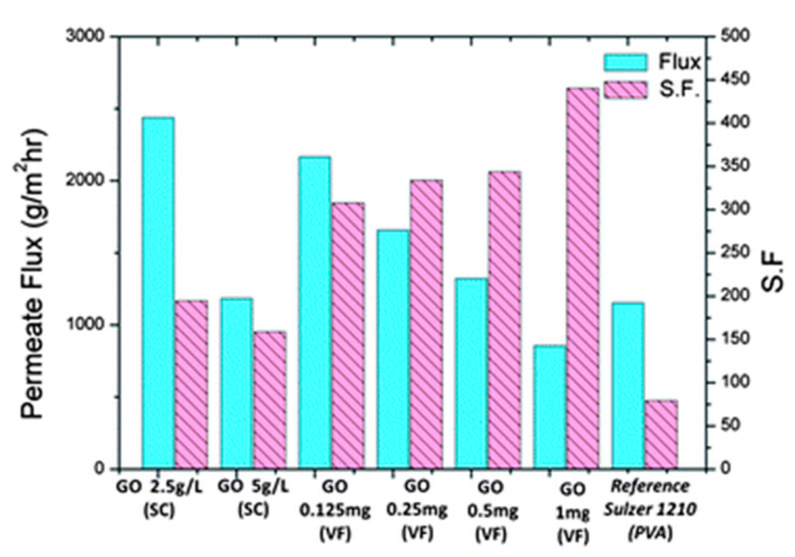
Permeate flux and separation factor (SF) brought about by the dehydration of ethanol via pervaporation using an 80 wt% ethanol feed solution at 70 °C.

**Figure 18 membranes-13-00108-f018:**
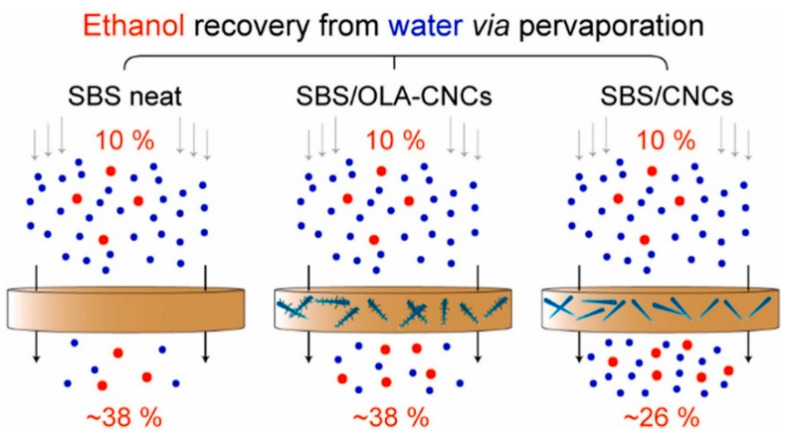
Performance of SBS/CNC and SBS/OLA-CNC nanocomposite membrane for ethanol separation.

**Figure 19 membranes-13-00108-f019:**
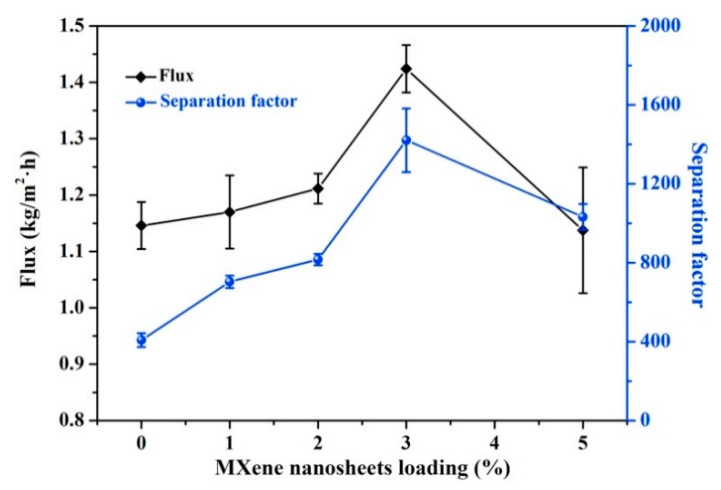
MXene loading effect on the pervaporation performance of MXene/CS MMMs for dehydration of 90 wt% water/ethanol mixtures at 50 °C.

**Figure 20 membranes-13-00108-f020:**
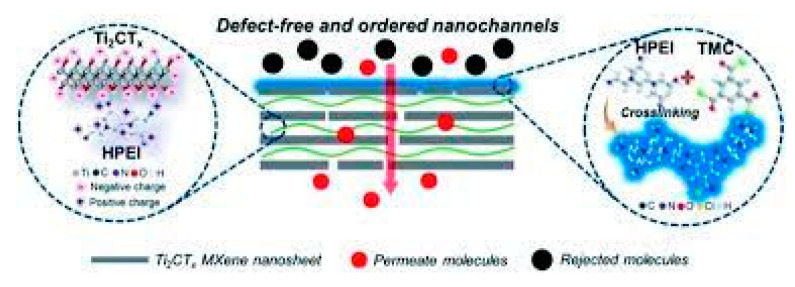
Schematic representation the ordered and fault-free MXene nanochannels produced by HPEI molecules following interfacial polymerization with TMC.

**Figure 21 membranes-13-00108-f021:**
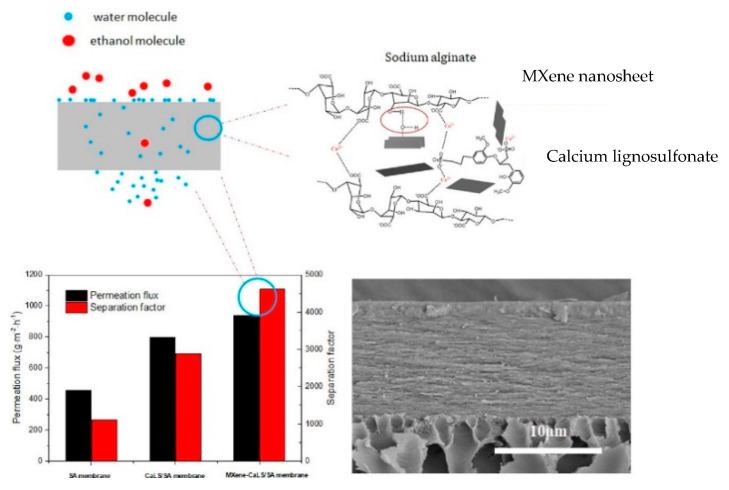
Hydrophilic calcium lignosulfonate combined with 2D-MXene, jointly modified the polysaccharide sodium alginate.

**Figure 22 membranes-13-00108-f022:**
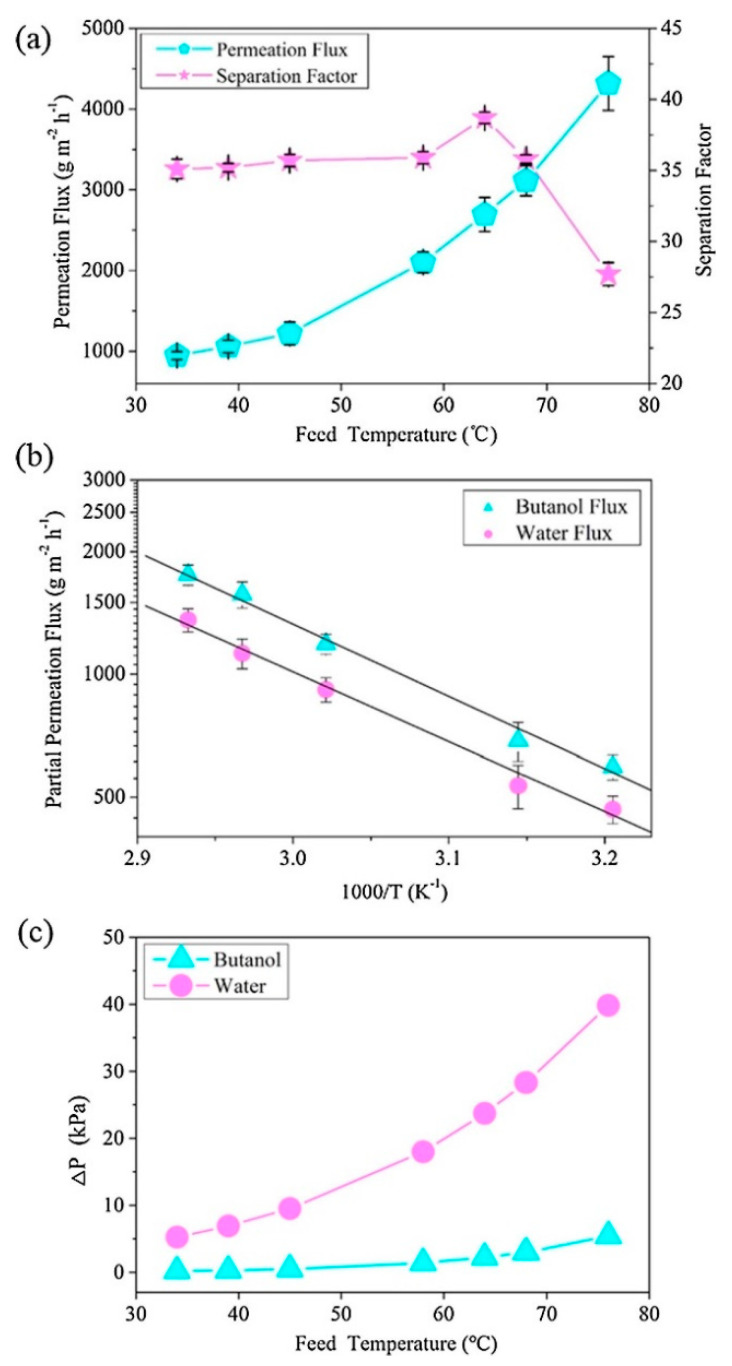
(**a**) Effects of feed temperature on permeation flux and separation factor of MMMs (COF_LZU1 loading of 1 wt%, separation layer thickness of 21 μm). (**b**) Arrhenius plots of partial fluxes. (**c**) Effect of temperature on the partial pressure difference of n-butanol and water. Effects of feed temperature on permeation flux and separation factor of MMMs (COF-LZU1 loading of 1 wt%, separation layer thickness of 21 μm).

**Figure 23 membranes-13-00108-f023:**
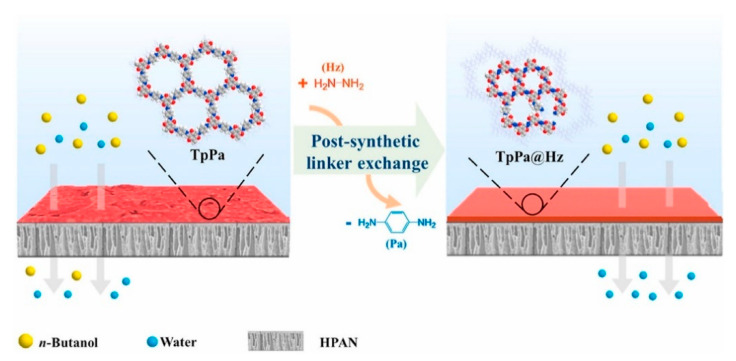
Schematic of post synthetic linker exchange method for the fabrication of COF membranes.

**Figure 24 membranes-13-00108-f024:**
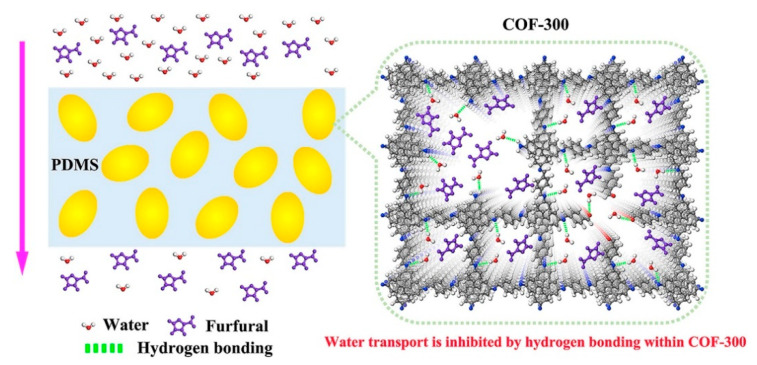
Schematic illustrating how COF-300 increases organic permeability while concurrently restricting water transport to improve pervaporation efficacy.

**Figure 25 membranes-13-00108-f025:**
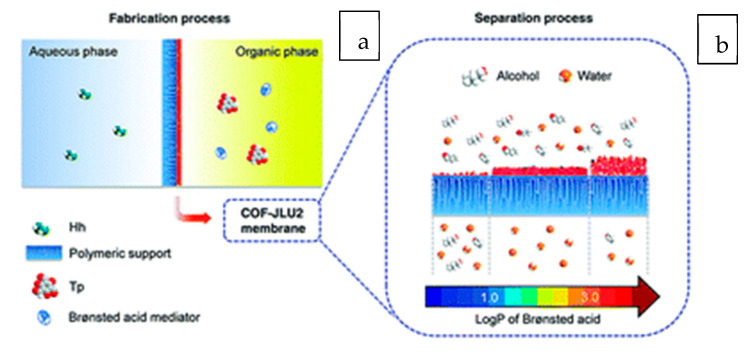
(**a**) Schematic illustration of Brønsted acid mediated one-step interfacial polymerization to fabricate a COF-JLU2 membrane on a polymeric support. (**b**) Reaction scheme of COF-JLU2.

**Figure 26 membranes-13-00108-f026:**
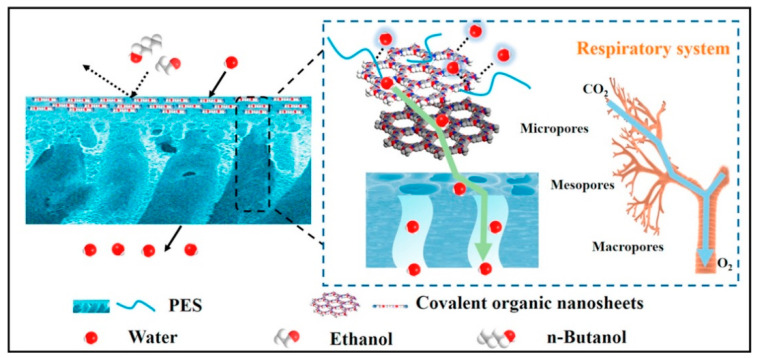
Schematic demonstrating effective water/alcohol separation using hierarchical pore structures made from 2D covalent organic nanosheets.

**Figure 27 membranes-13-00108-f027:**
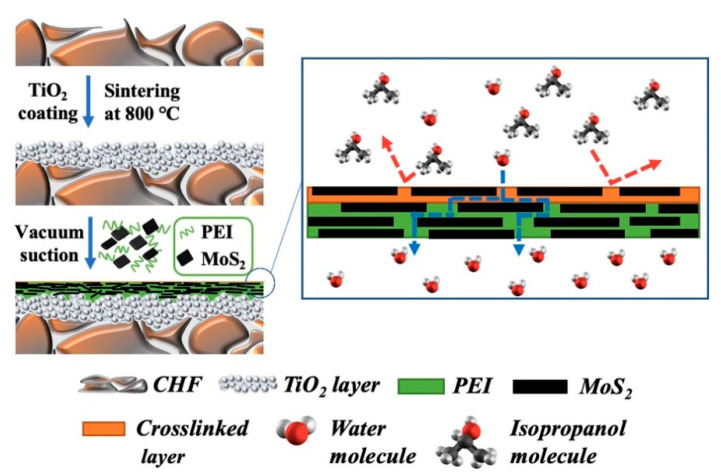
Schematic showing the arrangement of MoS_2_ hybrid membranes atop ceramic hollow fibers for effective pervaporation-based dehydration of isopropanol solutions.

**Figure 28 membranes-13-00108-f028:**
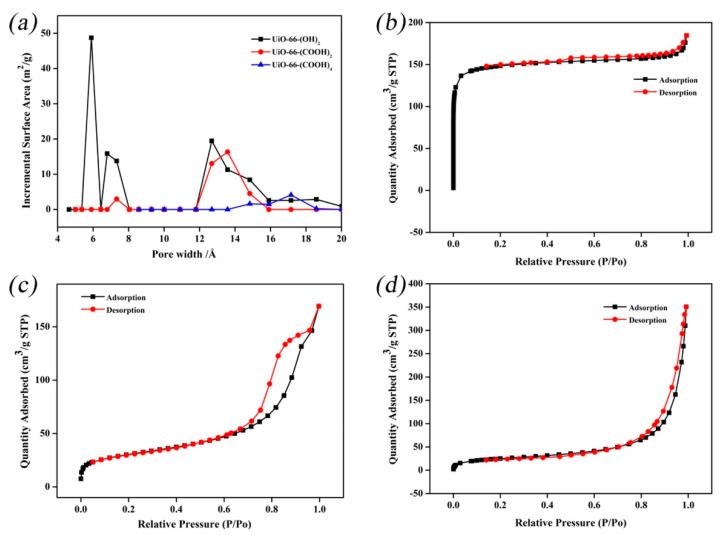
(**a**) Pore size distribution of UiO_66_X nanoparticles; N_2_ adsorption-desorption isotherms: (**b**) UiO_66_(OH)_2_; (**c**) UiO_66_(COOH)_2_; (**d**) UiO_66_(COOH)_4_.

**Table 1 membranes-13-00108-t001:** Comparison of various solvent recovery processes.

Technology	Advantages	Disadvantages	References
Adsorption	Cheaper Low to no energy required High efficiency High selectivity	High retention time Chemical addition required for regeneration Desorption of organic compound previously adsorbed on the sorbent is cumbersome Additionally, bacteria can adhere to the adsorbent and decrease the adsorption efficiency, especially if the adsorbent is recycled	[17,18]
Gas Stripping	The gas-stripping process has a number of benefits over other removal methods, including simplicity, cheap operating costs, and the absence of fouling or clogging brought on by the presence of biomass. Additionally, CO_2_ and H_2_ gases created during the fermentation process can be utilized for solvent recovery using gas stripping. Because only volatile products are removed from the fermentation broth, the chemical intermediates are mostly transformed into valuable compounds while remaining in the fermentation broth.	In a bioreactor, small bubbles created during gas stripping produce an excessive amount of foam. An antifoam agent must be added as a result of this procedure, and this chemical may be hazardous to microorganisms. This thus has the effect of lowering fermentation productivity overall.	[19,20,21]
Liquid-Liquid Extraction	Simple to Use Wide range of solvents to choose from	Tediousness Need to use large volumes of solvents of high purity, which can increase environmental pollution Often small enrichment factor of the analyte Low selectivity Formation of emulsions that are difficult to break Problems with handling samples of large volume	[22]
Pertraction	The pertraction method’s main benefit is the elimination of the need for extractant dispersion in the solvent phase. It is feasible to link certain membrane features with extractant capability via membrane pertraction.	Pertraction has various drawbacks, including less stable hollow fiber modules when in contact with solvent and poorer mass transfer coefficients compared to liquid-liquid extraction. Due to the relatively high viscosity of extractants, membrane solvent extraction issues on an industrial scale may arise. Due to these issues, the solvent phase experienced pressure losses and mass transfer restrictions.	[23,24]
Distillation	High alcohol recovery–With 99+% of the alcohol ending up in the alcohol product stream, distillation results in a low alcohol content in the treated bottoms stream. High concentration factor–utilizes many advantageous vapor-liquid equilibrium (VLE) phases that occur at low alcohol concentrations. At modest feed concentrations, adequate energy efficiencies Scales up well	Azeotropes–cannot satisfy the requirements for product dryness without adding additional separation processes, such as molecular sieve adsorption, or changing the process parameters. At low alcohol feed concentrations, energy requirements are noticeably higher. Operates at temperatures that are higher than standard fermentor optimal temperatures, often at temperatures that are fatal to microorganisms and that result in the inactivation of proteins and enzymes (unless under vacuum).	[2,25,26]
Membrane Separation	High efficiency Large number of separation needs can be met High selectivity	High energy required Fouling Scale up complexity	[27,28,29,30]

## Data Availability

Not applicable.

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
