# Peer review of "Recent Advancements in the Recovery and Reuse of Organic Solvents Using Novel Nanomaterial-Based Membranes for Renewable Energy Applications"

_membranes, 2023, doi:10.3390/membranes13010108_

Round 1

Reviewer 1 Report

Title: Recent Advancements in the Recovery and Reuse of Organic Solvents using novel nanomaterial-based membranes for renewable energy applications

Article Type: Review article

Manuscript Number: membranes-2123330-V1

This review article presents a membrane-based technologies using novel materials can improve the separation performance of organic solvents is considered. The technical methods in previous studies are discussed with the goal of emphasizing benefits and problems faced in order to direct research towards an optimized membrane separation performance for renewable fuel production on a commercial scale.

My recommendation is that the authors carefully consider the below points, revise appropriately.

1. The authors should consider more representative word in the keywords as significance as possible.

2. The authors may consider using more clear words or line in all figures as well as consistency symbol and color for legibility.

3. My suggestion is that the authors may use suitable software to draw the chemical formula as well as chemical equation and the text size is consistent with the text in this article. (Such as figure 6.)

4. My suggestion is that all chemical structures in this article should be expanding to enough scale for easy to see the chemical structure as well as the words in text. And the diagram in each figure should be suitable arrange in the center of figure as necessary as possible.

Author Response

Please find the attachment below.

Reviewer 2 Report

In this work, the authors reviewed how membrane-based technologies by using novel materials to improve the separation performance of organic solvents. The organization and discussion of this manuscript are logical and reasonable. In my opinion, it could be accepted after a moderate revision. Some suggestions or concerns are as follows,

1.     It would be better if the authors could use a table to summarize the advantages and disadvantages of different technologies for preventing solvent inhibition in the introduction section. Some related references to membrane fouling could be considered to cite to strengthen the background/discussion of the disadvantages of membrane technology in this table. For example, Journal of Membrane Science 565 (2018): 293-302; Journal of Colloid and Interface Science 613 (2022): 426-434; Environmental Science: Nano 6, 10 (2019): 3080-3090; Journal of colloid and interface science 517 (2018): 155-165.

2.     One Figure which could reflect the publication trend of membrane fabrication for organic solvent separation is strongly recommended.

3.     The resolution of all Figures in this manuscript is low.

4.      The authors focused more on fabricating different nanomaterial-related membranes and their separation performance for organic solvents. How about their disadvantages? Such as high/low permeability, leaching, low membrane stability, etc. How about the future directions to solve those issues? This could be mentioned in each section or added a new section before the conclusion.

Author Response

Please find the attachment below

Round 2

Reviewer 2 Report

I am happy with this version, and it can be accepted now.